# A Brief Overview: Sex Differences in Prevalent Chronic Musculoskeletal Conditions

**DOI:** 10.3390/ijerph20054521

**Published:** 2023-03-03

**Authors:** Demario S. Overstreet, Larissa J. Strath, Mackensie Jordan, Indonesia A. Jordan, Joanna M. Hobson, Michael A. Owens, Adrian C. Williams, Robert R. Edwards, Samantha M. Meints

**Affiliations:** 1Department of Anesthesiology, Perioperative and Pain Medicine, Brigham and Woman’s Hospital, Boston, MA 02115, USA; 2Harvard Medical School, Boston, MA 02115, USA; 3Department of Community Dentistry and Behavioral Science, University of Florida, Gainesville, FL 32603, USA; 4Pain Research and Intervention Center of Excellence (PRICE), University of Florida, Gainesville, FL 32610, USA; 5Amplify Health and Wellness, Newmarket, ON L3Y 8C6, Canada; 6Department of Psychology, University of Alabama at Birmingham, Birmingham, AL 35294, USA; 7Department of Psychiatry and Behavioral Sciences, School of Medicine, Johns Hopkins University, Baltimore, MD 21205, USA

**Keywords:** chronic pain, sex differences, musculoskeletal disorders

## Abstract

Musculoskeletal (MSK) pain disorders are some of the most prevalent and disabling chronic pain conditions worldwide. These chronic conditions have a considerable impact on the quality of life of individuals, families, communities, and healthcare systems. Unfortunately, the burden of MSK pain disorders does not fall equally across the sexes. Females consistently demonstrate more prevalent and severe clinical presentations of MSK disorders, and this disparity increases in magnitude with age. The aim of the present article is to review recent studies that have examined sex differences between males and females in four of the most common MSK pain disorders: neck pain, low back pain, osteoarthritis, and rheumatoid arthritis.

## 1. Introduction

Musculoskeletal diseases are the leading cause of both chronic pain and disability worldwide [1]. In fact, it is estimated that approximately 1.71 billion people are living with a musculoskeletal condition [2]. Notably, low back pain, neck pain, osteoarthritis, and rheumatoid arthritis are among the most disabling musculoskeletal diseases [3]. Musculoskeletal diseases affect bones, muscle, ligaments, joints, tendons, and nerves. They are often characterized by pain (acute or chronic) and they limit movement and dexterity, impacting people’s ability to function in society [1]. Musculoskeletal pain can affect anyone regardless of demographic characteristics; however, there are clear disparities between demographic groups whereby the burden of the pain prevalence and experience is extremely unequal. In sum, previous studies suggest that females are disproportionately burdened by these painful conditions and present with more persistent pain symptoms than their male counterparts [4]. The mechanisms underlying these sex differences are still being elucidated, though it is hypothesized that differences in hormones, immune system functioning and differences in the perception of pain are primary contributors [5]. Physiological and behavioral differences in pain responses between males and females have also been well documented [6,7].

It should be noted that in this article, sex is defined as the biological processes that differ between males and females based on genetics and hormones, and not gender—the psychosocial construction of oneself as a man or a woman—in all cases. Thus, the primary objective of this topical review is to highlight recent studies that have examined differences between the biological sexes in the four most common musculoskeletal conditions—neck pain, chronic low back pain, osteoarthritis and rheumatoid arthritis (Figure 1) [8,9,10,11].

### 1.1. Neck Pain

Neck pain (i.e., pain originating from/around cervical vertebrae 1–7) has become an increasingly common problem around the world. This is particularly exacerbated with the societal shift to working with technology and sitting at desks on computers, which is a primary risk factor for the development of persistent and disabling neck pain [12]. Neck pain has considerable impact on individuals, families, communities, and health care systems. Neck pain has an annual prevalence rate of over 30% in the United States, with nearly 50% of individuals experiencing some form of neck pain in their lifetime, and up to 67% of those individuals having pain that significantly detracts from their daily lives [13,14]. The etiology of neck pain is complex and believed to be a biopsychosocial problem, with the onset and course of neck pain being highly influenced by environmental and personal factors, some of which are modifiable. To better understand how multiple factors may contribute to increased risk of chronic neck pain, each domain of the biopsychosocial model must be investigated in order to predict and help prevent long-term neck pain. Having a history of acute neck or low back pain, poor self-assessed health and poor physiological status have been found to increase the risk of chronic neck pain onset [15]. In the physical domain, job-related exposure, routine physical activity and impairments in cervicoscapular strength, mobility and endurance may be associated with neck pain [16]. In the neurophysiological domain, evidence has shown alterations in pain processing that result in localized and widespread hypersensitivity to mechanical stimuli, as well as impairments in diffuse noxious inhibitory control (DNIC) are found in individuals with persistent pain [16]. Lastly, psychosocial factors including job satisfaction and general measures of psychological health have been shown to predict long-term symptoms and disability associated with neck pain. 

In general, the prevalence of neck pain, regardless of source, is higher in females [15]. Data suggest that these differences in idiopathic neck pain may be due to muscle stiffness or alterations in neuromuscular control [17], though there is no consistent evidence suggesting one pathology over another. In 2016, a 5 year cross-sectional, observational study in Sweden was able to establish sex-related prognostic factors that show less favorable recovery in the female sex after neck trauma. Females were found to be twice as likely to have persistent pain after neck trauma likely due to higher vulnerability to tissue damage based on structural factors, higher risk for psychological distress, the pain sensitization process and possibly social factors [18]. Most studies indicate a higher incidence of neck pain among females and an increased risk of developing neck pain until the 35–49 year age group, after which the risk begins to decline [15,19]. However, it should be noted that these sex differences are also observed in younger individuals, with female children and adolescents reporting that they experienced pain more often in the past 7 days compared to males of their age [20].

Anatomically, females have smaller cervical vertebrae resulting in less segmental support area (disk and facet joints), less muscle strength, and ligament stiffness resulting in increased pain, reduced stability and increased range of motion, lower tolerance limit for lower neck shear force, and faster muscle reaction times resulting in greater tissue strain and injury potential [18]. Approximately 10% of neck pain cases are associated with conditions such as polymyalgia rheumatica, fibromyalgia, and rheumatoid arthritis, all of which are more prevalent in the female population [19,21,22,23]. In migraine, another female-predominant pain condition, cutaneous allodynia, cervical mobility, muscle function and overall migraine severity was greater in individuals who also demonstrated the presence of chronic neck pain [24]. Additionally, it has been reported that people who suffer from chronic neck pain are three times more likely to have chronic low back pain than individuals from the general population [25]. 

### 1.2. Low Back Pain 

Low back Pain (LBP) is the most prevalent musculoskeletal condition in America and has been the single leading cause of disability worldwide since 1990 [26]. According to the Global Burden of Disease, the prevalence of LBP increased from 377.5 million in 1990 to 577 million in 2017 and continues to increase as the overall population ages [27]. Though the risk of developing LBP increases with age until peaking in the 7th and 8th decades of life [28], individuals between the ages of 50 and 54 constitute the largest group of people currently living with LBP [26]. It is estimated that 50–80% of the population will experience low back pain at some point in their lifetime [29]. While the long-term prognosis is favorable for most people who experience acute LBP, roughly one quarter of these individuals will go on to develop chronic low back pain [30,31]. Persistent or chronic low back pain (cLBP) is defined as pain that occurs in the lower back and persists for a period of 12 weeks or longer, and the overwhelming majority (~90%) of cLBP is “non-specific” and is not attributable to any distinguishable pathology associated with the anatomical structures positioned between the posterior margin of the ribcage and the gluteal fold (lower back) [32]. 

While cLBP can affect anyone regardless of biological sex, the prevalence is higher in females across all age groups. Similar to neck pain, a growing body of chronic pain studies endorses a biopsychosocial model of pain which posits that pain perception in cLBP is attributable to factors that are biological and psychosocial in nature [33]. It is well known that psychological distress contributes to the experience of cLBP and increases the risk of associated disability [34]. Depression is one of the strongest predictors of cLBP and females are nearly twice as likely to be diagnosed than males [35]. Further, depression is estimated to be 3–4 times greater in chronic pain patients than in the general population, of which females carry the burden [4]. Thus, early identification of depression and treatment of depressive symptoms might provide supplemental benefits to females who are suffering from cLBP. Anxiety has also been linked to chronic low back pain [36]. As in the case of depression, anxiety disorders and symptoms are twice as likely to occur in females than males [37], though more research is needed to determine whether anxiety moderates the relationship between biological sex and cLBP outcomes. 

Aside from psychological correlates associated with low back pain, physiological responses to pregnancy, physical and emotional stress related to child-rearing, and perimenopausal weight gain in the abdominal area are also directly linked to this chronic condition [38]. In postmenopausal females, lumbar discs are susceptible to accelerated degeneration compared to age-matched males, further increasing the risk of developing cLBP for females over the age of 50 years [38]. In addition to these physiological factors, genetics might also contribute to the development of cLBP in females [28]. In a relatively large (*n* = 2256) cross-sectional study involving identical and fraternal twins, it was determined that genetic background and lumbar disc degeneration were the primary risk factors for severe and disabling LBC in this all-female sample [39,40]. More recently, a 2021 study revealed sex-specific epigenetic signatures in T cells that differentiated cLBP patients from pain-free controls. Female subjects with cLBP presented with hypomethylated expression in genes that have functions in the immune system [41]. A greater concentration of immune markers such as pro-inflammatory cytokines have been reported in cLBP [42] much like other MSK disorders such as osteoarthritic disease [43].

### 1.3. Osteoarthritis

Osteoarthritis (OA) is a widespread musculoskeletal condition, affecting approximately 300 million individuals worldwide. It is the leading cause of disability in older adults, with an estimated 10%–15% of individuals over the age of 60 having some degree of OA in one or more joints. OA is characterized by the degeneration of cartilage and bone as well as bony overgrowth (bone spurs), which eventually leads to pain, stiffness, and loss of function in the joint. The joints most commonly affected are those in the knees, hips, carpals and metacarpals, as well as the spine. OA is normally brought about by injury, overuse, and repetitive stress to the areas, but genetics and anatomy also play a distinct role. 

Population-based studies have revealed that OA prevalence is higher in females compared to males, and that females generally have a higher risk of developing OA across the lifespan, especially after menopause [44]. Additionally, females over the age of 55 also tend to have more severe OA in the knee [45]. Interestingly, females are three times less likely to undergo hip or knee arthroplasty than males. In experiments aimed at understanding sex differences in pain severity of OA, results indicate that females report higher pain severity via visual analogue scale, greater prevalence of pain in both knees compared to just one knee, greater levels of inflammation, and more impaired function compared to males. In females but not males, low serum levels of endogenous estradiol, progesterone and testosterone were associated with increased pain, synovial inflammation, and decreased cartilage volume [46]. When analyzing synovial fluid from the knees of OA patients, the balance of pro- and anti-inflammatory factors appeared to differ as a function of sex; specifically, the anti-inflammatory chemokines MMP-10, IL-8, CCL-4, and monocyte chemoattractant protein (MCP)-2) were higher in males, whereas proinflammatory cytokines (IL-6, IL-10, IL-1β, TNF-α) were higher in females [47,48]. Such sex differences in inflammatory activity may contribute to the amplified pain sensitivity observed in females (compared to males) with osteoarthritis. Specifically, females with OA are more pain-sensitive on laboratory measures of pain sensitivity, and also demonstrate enhanced reactivity of IL-6 and blunted diurnal rhythms of anti-inflammatory neuroendocrine factors such as cortisol [49,50].

Psychosocial measures such as depression, anxiety, and social support, as well as physical activity appear to be similar between the sexes in the case of OA [48,51,52]. One study [53] used a single-blind placebo design to evaluate the placebo response to pain, depressive and anxiety symptoms, and performance-based tests in patients with knee OA. Results showed that males reported greater depressive symptoms due to their OA than females, while females showed greater walking resistance due to their OA than men.

### 1.4. Rheumatoid Arthritis

Rheumatoid arthritis, or RA, has many of the same symptoms as OA including pain and stiffness in the joints; however, their underlying pathologies differ. RA, as opposed to OA, is an autoimmune disorder in which the immune system attacks cells in the joints, causing inflammation, swelling, and pain at the affected areas [54]. RA commonly affects the tissue in the hands, wrists and knees. As this disease and subsequent tissue damage is long-lasting, it typically leads to chronic pain, lack of physical function, and deformity of the affected joints [55]. RA can also exert its effects extra-articularly on visceral organs, usually targeting the lungs, heart and eyes [56]. In addition to pain and stiffness, other symptoms such as fever, fatigue, weakness and weight loss can also accompany RA [57]. 

Many autoimmune disorders, including RA, display a disproportionate burden between the sexes, whereby females are afflicted at greater rates than males [58,59]. Research has documented sex differences in RA across several domains including epidemiology, disease-course, and management, leading the experience of the disease to differ between males and females [60]. Typically, females experience a more deleterious disease course, showing greater amounts of disease activity as well as prevalence of RA. These differences are what lead females to have poorer treatment outcomes compared to males with the same disorder. The reasons underlying this difference remains unclear, though it is thought that X-linked genetic factors [61], hormones, and immune system differences may be involved. In a study by Yu et al., the authors concluded that differences in expression of interleukin (IL)-4, a pro-inflammatory cytokine that participates in cytokine–cytokine receptor interactions, T-helper cell differentiation, and T cell receptor signaling pathway (among other things) between the sexes may contribute to the molecular mechanism of sex differences in RA [62]. Better disease activity scores in males may also be attributable to phenotypic factors such as muscle strength and bone density, which typically is greater in males compared to females [63]. These differences may allow for better compensatory strategies for males, making their daily activities less affected by RA. Because pharmaceutical treatment options do not differ by sex, this has likely led to decreased levels of adherence to RA-specific medications in females of child-bearing age. Because many of the drugs used to treat RA, such as methotrexate, can be teratogenic and passed along from mother to child, there is no current safe dosage recommended for females planning to conceive and breast feed [64,65]. Psychological factors associated with worse pain and RA outcomes, such as anxiety and depression, are also more common in females, and could also partially explain the sex differences seen in RA [66]. 

## 2. Conclusions

Musculoskeletal conditions, namely neck pain, back pain, osteoarthritis, and rheumatoid arthritis, are highly prevalent and impact females more frequently and more severely than males. We have highlighted biopsychosocial factors associated with MSK pain and more specifically, the mechanistic factors that may be contributing to sex differences in MSK pain including genetics, immunology, and hormones. By better understanding these mechanisms, we can develop targeted treatments to help ameliorate sex disparities in MSK pain. It is important to note that this review focused solely on sex as a biological variable and as such has not addressed the influence of gender, a social construct, which also has considerable effects on the pain experience. Future research should examine the intersectionality of sex and gender on pain.

## Figures and Tables

**Figure 1 ijerph-20-04521-f001:**
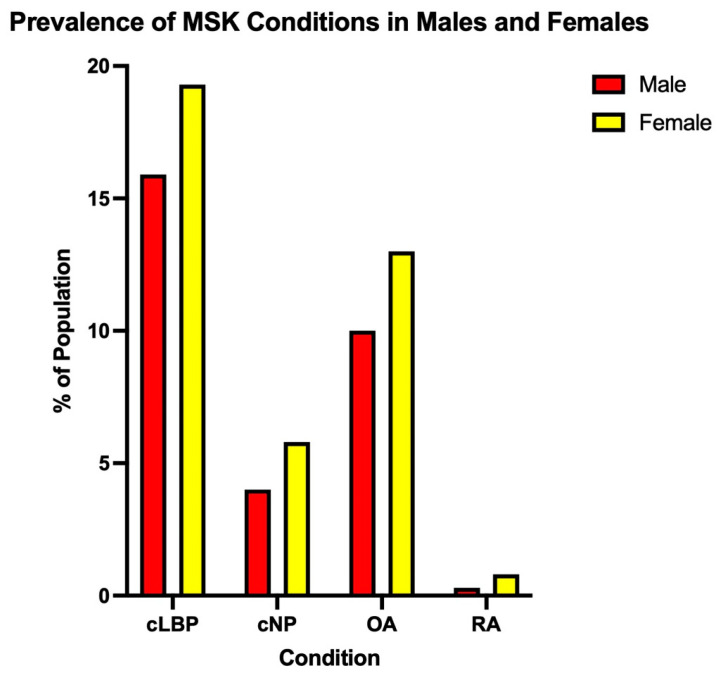
Prevalence of the top four musculoskeletal conditions in the United States (Adapted from Kahere et al. (2021), Wang et al. (2016), Zhang et al. (2010) and Gabriel (2001)) [8,9,10,11].

## Data Availability

No new data were created/collected.

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
