# Peer review of "A Brief Overview: Sex Differences in Prevalent Chronic Musculoskeletal Conditions"

_ijerph, 2023, doi:10.3390/ijerph20054521_

Round 1

Reviewer 1 Report

Excellent, No conclusion.

The main questions addressed by this manuscript is biological sex differences in musculskeletal pain conditions. A review of the evidence. This is relevant and interesting. These are the commonest sources of pain. Pain has a huge impact on individuals and society. A review of evidence is worthwhile. This has not being covered before. It is well written and easy to read. Conclusion part could be improved. Limitations discussed.

Author Response

We would like to thank the editors and reviewers for their careful and thoughtful comments on our work. In each case, we have shown the comments below and our responses as bulleted points. Where needed, changes to the manuscript have been identified and revised in blue font. We believe that the changes warranted by the Reviewers’ comments have made our manuscript stronger as a result and hope to hear from you soon regarding our manuscript’s suitability for the International Journal of Environmental and Public Health.

 Thank you very much for your consideration.  We hope these changes are satisfactory.

Reviewer 1:

--The main questions addressed by this manuscript is biological sex differences in musculoskeletal pain conditions. A review of the evidence. This is relevant and interesting. These are the commonest sources of pain. Pain has a huge impact on individuals and society. A review of evidence is worthwhile. This has not being covered before. It is well written and easy to read. Conclusion part could be improved. Limitations discussed.

--We would like to thank the reviewer for their enthusiasm for our work, and for catching this error.  We have added a conclusion section to the manuscript on pages 10-11.

Reviewer 2 Report

Contributions are not well written, so please correct that and specify who has done what. 

Author Response

We would like to thank the editors and reviewers for their careful and thoughtful comments on our work. In each case, we have shown the comments below and our responses as bulleted points. Where needed, changes to the manuscript have been identified and revised in blue font. We believe that the changes warranted by the Reviewers’ comments have made our manuscript stronger as a result and hope to hear from you soon regarding our manuscript’s suitability for the International Journal of Environmental and Public Health.

 Thank you very much for your consideration.  We hope these changes are satisfactory.

Reviewer 2:

--Contributions are not well written, so please correct that and specify who has done what. 

--Thank you for noticing this.  We have added a detailed author statement at the end of the document prior to the references (page 11).

Reviewer 3 Report

Reviewer comments

Journal

Environmental research and public health

Title

A Topical Review: Biological Sex Differences in Prevalent Chronic Musculoskeletal Conditions

The manuscript covers an important topic which is the sex differences in musculoskeletal condition

The authors need to address some points:

21… The aim of the present manuscript…the aim of this article

30.. They impair the 30 functions of bones…. what is meant by impair the function of bones

42…. that in this manuscript sex…the word “manuscript” was repeated multiple times ..which is not the appropriate word

45.. topical review is highlight…English editing is required in the whole manuscript

78… Most studies indicate…the authors should indicate these studies as only one reference was cited

The manuscript does not add any new to the already present literature

The authors should determine the research gap clearly

The authors should determine the pathophysiology of this difference and if there any hormonal cause for that

The review needs more graphs or tables to illustrate the topic .

Author Response

We would like to thank the editors and reviewers for their careful and thoughtful comments on our work. In each case, we have shown the comments below and our responses as bulleted points. Where needed, changes to the manuscript have been identified and revised in blue font. We believe that the changes warranted by the Reviewers’ comments have made our manuscript stronger as a result and hope to hear from you soon regarding our manuscript’s suitability for the International Journal of Environmental and Public Health.

 Thank you very much for your consideration.  We hope these changes are satisfactory.

Reviewer #3:

21… The aim of the present manuscript… The aim of this article

-We have revised the manuscript to reflect the advice above. The term “manuscript” has been replaced with “article” throughout the paper.

30… They impair the 30 function of bones… what is meant by impair the function of bones.

  • We agree that this was unclear and have revised the sentence as follows: “ “These conditions affect bones, muscles, ligaments, joints, tendons, and even nerves.”

42… that in the manuscript sex…the word “manuscript” was repeated multiple times… which is not the appropriate word.

  • We agree with the reviewer and have revised the document to accommodate the request above. The word “manuscript” has now been replaced with “article”.

45… Topical review is highlight… English editing is required in the whole manuscript.

  • We have added the word “to” to the sentence referenced above (page 3). The document now reads “Thus, the primary objective of this topical review is to highlight recent studies that have examined differences in the pain experience between the biological sexes,’’. Further, the manuscript has been reviewed and edited by the team, including an additional co-author Dr. Robert Edwards, who has published extensively in the area of pain science.

78…  Most studies indicate… the authors should indicate these studies as only one reference was cited.

  • We thank the reviewer for their critique. The article referenced was an epidemiological review article, but as per the reviewer’s request, a more recent epidemiological report/review was referenced on page 5. 

---       The manuscript does not add any new to the already present literature.

--The purpose of this manuscript is to provide the readership with an overview and synthesis of recent studies that address sex differences in the most common musculoskeletal disorders.  Currently, there is no such review in the pre-existing literature.

--- The authors should determine the research gap clearly.

--We have updated the statement for clarity of the research gap: “The mechanisms underlying these sex differences are still being elucidated, though it is hypothesized that differences in hormones, immune system functioning and differences in the perception of pain are primary contributors.” (Page 3, Lines 54-56).

---The authors should determine the pathophysiology of this difference and if there are any hormonal cause for that.

--Throughout the manuscript, authors have described the unique pathophysiology found to be associated with each musculoskeletal condition. Hormonal causes were identified as key contributors for the conditions that have strong evidence to support this claim. For example, on page 8, line 170, we highlight how low hormone levels in females are associated with worse pain and disease markers in OA.  

---The review needs more graphs or tables to illustrate the topic.

We would like to thank the reviewer for their comment.  A figure illustrating the differences in prevalence of the musculoskeletal conditions has been added to the manuscript (see page 3, line 64).

Round 2

Reviewer 3 Report

The manuscript has been improved to a great extent